# On the relationship between emotions and cognitive control: Evidence from an observational study on emotional priming Stroop task

**Antonino Visalli**[1]*, **Ettore Ambrosini**[2,3,4], **Giada Viviani**[2,3], **Fabio Sambataro**[2,3], **Elena Tenconi**[2,3], **Antonino Vallesi**[2,3]*

**1** IRCCS San Camillo Hospital, Venice, Italy, **2** Department of Neuroscience, University of Padova, Padova, Italy, **3** Padova Neuroscience Center, University of Padova, Padova, Italy, **4** Department of General Psychology, University of Padova, Padova, Italy

* antonino.visalli.av@gmail.com (AV); antonino.vallesi@unipd.it (AV)

**Editor:** Valerio Manippa, University of Bari Department of Education Psychology and Communications: Universita degli Studi di Bari Aldo Moro Dipartimento di Scienze della Formazione Psicologia Comunicazione, ITALY

## Abstract

Evidence is discordant regarding how emotional processing and cognitive control interact to shape behavior. This observational study sought to examine this interaction by looking at the distinction between proactive and reactive modes of control and how they relate to emotional processing. Seventy-four healthy participants performed an emotional priming Stroop task. On each trial, target stimuli of a spatial Stroop task were preceded by sad or neutral facial expressions, providing two emotional conditions. To manipulate the requirement of both proactive and reactive control, the proportion of congruent trials (PC) was varied at the list-wide (LWPC) and item-specific (ISPC) levels, respectively. We found that sad priming led to behavioral costs only in trials with low proactive and reactive cognitive control demands. Our findings suggest that emotional processing affects cognitive processes other than cognitive control in the Stroop task. Moreover, both proactive and reactive control modes seem effective in overcoming emotional interference of priming stimuli.

## Introduction

Cognitive control refers to the ability to adaptively regulate information processing and behavior according to current goals [1, 2]. A core function of cognitive control is conflict resolution, that is, fostering task-relevant information processing and/or response selection, whilst ignoring conflicting/distracting irrelevant information and/or inhibiting prepotent responses [3, 4]. A particular set of stimuli that may be regarded as a source of interference during cognitive control tasks is represented by emotional stimuli. Due to their inherent survival value, indeed, emotional stimuli should be able to automatically withdraw informational processing resources from ongoing cognitive tasks or at least compete for common-pool resources [5]. It follows that, although emotional processing can be adaptive in some situations (e.g., in the presence of threats), it may represent a source of interference when emotional stimuli are not

**Data Availability Statement:** Data and analysis codes are available at the Open Science Framework: osf.io/bc9x3/.

**Funding:** This study was in part supported by the "Department of Excellence 2018-2022" initiative of the Italian Ministry of University and Research (MIUR), awarded to the Department of Neuroscience – University of Padua, by "Progetto giovani ricercatori" grants from the Italian Ministry of Health (project code: GR-2018-12367927 – FINAGE, to A.Va.; project code: GR-2019-12371166, to E.A.), and the PRIN 2020 grant (protocol 2020529PCP) from the Italian Ministry of University and Research (MUR) to E.A. The funders had no role in study design, data collection and analysis, decision to publish, or preparation of the manuscript.

**Competing interests:** The authors have declared that no competing interests exist.

part of current goals. Despite the plausibility of these hypotheses, findings about the influence of emotional processing on cognitive control are often inconsistent across studies [6]. The present study sought to contribute to our understanding of the interaction between emotional processing and conflict resolution by taking into account an aspect of cognitive control rarely addressed in the emotion literature, that is, the distinction between proactive and reactive modes of control [2].

Following the influential dual-mechanisms of control (DMC) framework [2], cognitive control can be exerted through two distinct operating modes: proactive and reactive control. Proactive control is an early selection mechanism that actively maintains in memory the task-relevant information and anticipatorily biases attention, perception and action according to task goals. Proactive control exerts a anticipatory and sustained activity before the occurrence of conflict to facilitate the processing of the task-relevant information, in the face of conflicting task-irrelevant one. Reactive control, by contrast, is a late correction mechanism, triggered by conflict detection in a just-in-time manner, thus reflecting the transient reactivation of task goals.

A general conclusion from the studies investigating how proactive and reactive modes can relate to emotion regulation is that proactive control may be effectively used for reducing emotional distraction during perceptual tasks [7–11]. Here we considered the other side of the coin, that is, whether and how emotional processing influences proactive and reactive control modes during a cognitive control task. Emotions permeate our everyday life [12]. Moreover, emotional disturbances are a central feature of many psychopathological conditions [13–15] and are commonly associated with cognitive dysfunctions [16, 17]. Thus, investigating the influence of emotions on cognitive control is relevant to understand control processes in daily life and in clinical and subclinical populations.

To this aim, we capitalized on the Stroop task [18], one of the most widely used tasks to investigate conflict resolution [19] and the interaction between emotion and cognitive control [20]. In its original and most popular version, namely, the color-word Stroop task, individuals are required to name the ink color of words denoting color names. The Stroop effect is a universal cost [21, 22], which consists in longer response times (RTs) and lower accuracy in naming the ink color of words written with a different color (e.g., the word "blue" printed in red; incongruent trials) as compared to words in which the ink color and the word name match (e.g., "blue" printed in blue; congruent trials).

It is possible to identify three families of the emotional adaptations of the Stroop task [23]. The first one pertains to paradigms in which participants are required to name the ink color of emotional vs. neutral words (i.e., emotional Stroop task). In the second family, the Stroop stimulus is characterized by emotional words (either positively- vs. negatively-valenced words or emotional vs. neutral words) overlaid on faces expressing emotions congruent or incongruent with the superimposed word (i.e., word-face Stroop task). The last family is the emotional priming Stroop tasks, in which an emotional vs. neutral stimulus is presented prior to a non-emotional Stroop task. In the present study, we used the last described emotional adaptation of the Stroop task as the first two presented some methodological issues. Concerning the emotional Stroop task, it does not represent a proper Stroop task. In line with the dimensional overlap taxonomy proposed by Kornblum [24], a Stroop task should ensure a dimensional overlap both between task-relevant and task-irrelevant stimulus dimensions and between each of them and response dimensions, two conditions that are instead unfulfilled in the emotional Stroop task. In other terms, there is no conflict in the emotional Stroop task. In addition, lexical differences between emotional and control words represent a confound that is difficult to control for [25]. Concerning the word-face Stroop task, a conflict between task-relevant and task-irrelevant stimulus dimensions is present, but emotional and cognitive conflicts are

confounded. Therefore, it is not possible to accurately estimate the influence of emotional processing on cognitive control. Unlike these two tasks, the emotional priming Stroop task does not typically present the above-mentioned issues (i.e., incomplete dimensional overlap, and lexical and semantic confounds). Moreover, the use of priming stimuli (i.e., emotional stimuli that are detached from the main task) allows researchers to investigate how task-irrelevant emotional information interferes with ongoing cognitive control processes.

In the current study, we employed an emotional priming spatial Stroop task to investigate whether the processing of emotions interacts with cognitive control and with which mode (i.e., proactive and/or reactive control) this interaction would occur. The spatial version not only satisfies the requirements for being a proper Stroop task, but it also has several advantages over the original color-word Stroop task. Briefly, the use of spatial stimuli excludes linguistic processing, minimizing potential verbally-related confounding effects and promoting a domain-general investigation of cognitive control. By requiring manual responses, it is less prone to assessment errors [22, 26]. Moreover, this paradigm has been successfully used to investigate neural correlates of proactive cognitive control modulations [27]. Emotional priming stimuli were sad faces. Sadness is one of the most frequently experienced emotions in everyday life [12]. Here, it was selected not only for its life relevance, but also with the aim of conducting a further study with patients affected by depression. Cognitive control demands were varied by manipulating the proportion of congruency (PC), namely the proportion of congruent trials in a task. Indeed, in high PC conditions, conflict is less likely and cognitive control demand is lower, whereas, in low PC conditions, trials are mostly incongruent and cognitive control is required to a greater extent [28, 29]. To distinguish proactive and reactive control modes, the PC manipulation was implemented both in a list-wide (LWPC) and item-specific (ISPC) manner: the former manipulation allowed us to explore the anticipatory and sustained activation of proactive control, whereas the latter served to isolate the reactive control mechanism [29]. Typically, in the LWPC manipulation, the PC is varied at the block level to obtain blocks with high PC and blocks with low PC; the Stroop effect is expected to be smaller in blocks with low PC compared to blocks with high PC [28]. The ISPC manipulation, by contrast, entails varying the PC of the items within each block; in this case, the Stroop effect is reduced in low PC items, as compared with high PC ones [30]. Unlike previous studies, in which LWPC and ISPC were manipulated independently and tested in separate blocks [29], we simultaneously manipulated both of them. See the Methods section for details and rationale on these manipulation implementations.

Previous research between emotions and conflict resolution in the Stroop task has provided mixed results. Indeed, compared to emotional neutral stimuli, negative stimuli were found to improve (i.e., smaller Stroop effects), impair (i.e., larger Stroop effects), or have no significant effect on cognitive control [6]. Accordingly, it is challenging to formulate unique hypotheses that specify the expected relationship between emotions and cognitive control. Regarding the interaction between emotional priming and conflict resolution, two opposite predictions can be made. On the one hand, if processing of negative emotions competes for resources with concurrent cognitive processing, as suggested in previous studies [5], we should observe larger Stroop effects following sadness priming compared to neutral priming. On the other hand, if negative stimuli, specifically sad stimuli, narrow attention [31], hence, reducing conflict caused by irrelevant stimulus dimensions or distracting stimuli [6, 32], we should observe smaller Stroop effects following sadness priming (i.e., improved cognitive control for sad stimuli).

Concerning the interaction between emotional priming and proactive control, three different hypotheses can be made. If emotional stimuli share cognitive resources with cognitive control processes, we should observe a decrease in the expected modulation of the Stroop effect by proactive control. More specifically, the expected Stroop effect reduction with higher level of

proactive control (i.e., lower LWPC) should be less prominent with sad priming. This is because the elaboration of negative emotional priming stimuli should withhold resources necessary to exert proactive control [33, 34]. Alternatively, it has been suggested that negative emotions increase proactive control [35]. In case higher levels of proactive control are exerted, the reduction in the Stroop effect should then be more pronounced after sad priming. Additionally, it would be possible that high levels of proactive control might prevent negative emotional priming from engaging cognitive resources needed for conflict resolution [7]. In this scenario, the increase in the Stroop effect following sad priming would be expected to be smaller when higher levels of proactive control are exerted.

Finally, concerning the interaction between emotional priming and reactive control levels, two opposite hypotheses can be made. If emotional priming engages cognitive resources needed for exerting reactive control, we should observe a decrease in the expected modulation of the Stroop effect by reactive control. In other terms, we should observe a less prominent Stroop effect reduction with higher levels of reactive control (i.e. lower ISPC) after sad priming. Conversely, if reactive control overcomes emotional interference, the increase in the Stroop effect following sad priming should be reduced with higher levels of reactive control.

## Methods

### Participants

Seventy-four participants completed the task online between September 2021 and January 2022 (44 females, 28 males; mean age = 24.3 years, SD = 4.8; two participants did not provide age and sex information; 64 right-handed). All participants reported no current (or history of) neurological or psychiatric disorders and of not being under the influence of alcohol or other drugs that might affect cognitive functioning. Participants gave their informed consent to participate in the study, which was conducted in accordance with the ethics standards of the 2013 Declaration of Helsinki for human studies of the World Medical Association. The project was previously approved by the Ethical Committee for the Psychological Research of the University of Padova (approved protocol reference number: 4187). Authors did not have access to information that could identify individual participants during or after data collection, as the data were collected anonymously online (contact information was not a compulsory item).

The method introduced by Westfall and colleagues [36] was used to perform a power analysis for a fully-crossed linear mixed-effects model, assuming participant and stimulus intercepts, participant slope, and residual variance partitioning coefficients of .1, .1, .2, and .6, respectively, as estimated conservatively from some recent unpublished studies with a similar design from our research group. The other variance partitioning coefficients were set to 0, as those effects were not included in the models we tested. This analysis revealed that a sample size of 74 participants (with 40 stimuli, see below) was large enough to detect a small effect size (Cohen's d = .3) with a power of .80. It should be noted, however, that this approach is not fully adequate for complex mixed effect models like the one used in this work, but it nonetheless provides a useful estimation of the so-called minimal statistically detectable effect for our study (i.e., the lower bound of the range of effect sizes that can be detected). Indeed, to the best of our knowledge, to date there are no accepted analytical approaches to accurately compute statistical power for such models. To provide another estimate of our minimal statistically detectable effect, which could also facilitate comparison with future studies using more standard analytical approaches, we performed a sensitivity power analysis in G*Power [37] for a repeated measure ANOVA for the Congruency $\times$ LWPC $\times$ Emotion interaction (see below), assuming a correlation between repeated measures of .75. This analysis revealed that a sample size of 74 participants was large enough to detect a small effect size (d = .21, corresponding to

$\eta^2_{\text{p}} = .01$) with a power of .80. Still, it should be noted that G*Power (and, to the best of our knowledge, all other software commonly used to compute power) does not support power calculation for general linear model effects including both multiple within-subjects factors and continuous covariates.

### Procedures and material

Participants performed an adapted version of the perifoveal Stroop task [26] with emotional priming and changes of the probabilistic context (see below). The experiment was programmed using Psytoolkit [38, 39] and administered online. All the participants were recruited by the experimenters and given a link to perform the task online.

The task was executed in full screen mode. The stimuli were presented on a 1024 x 768 pixels grey background (RGB: 128, 128, 128). Each trial (Fig 1) started with a fixation stimulus, consisting of a thin black cross (30 x 30 pixels) enclosed in the partial outline of a black square (94 x 94 pixels), on which participants were instructed to keep their gaze. After 500 ms, the fixation stimulus was replaced by an emotional or neutral stimulus displayed at the center of the screen. The visual stimuli were selected from the NimStim set of facial expressions [40] and consisted in 20 photographs portraying sad emotional expressions and the corresponding 20 neutral expressions (see S1 Appendix for the list of used images). The sad images were selected such as to have a set balanced for sex (i.e., 10 female and 10 male portrayed actors) with highest validity and test-retest reliability as reported in the original work [40]. After 600 ms, the emotional prime was replaced by the fixation stimulus. After 400 ms, a target stimulus appeared at one of the four internal corners of the fixation stimulus (upper-left, upper-right, lower-right, and lower-left). The target stimulus consisted in an arrow pointing towards one of four

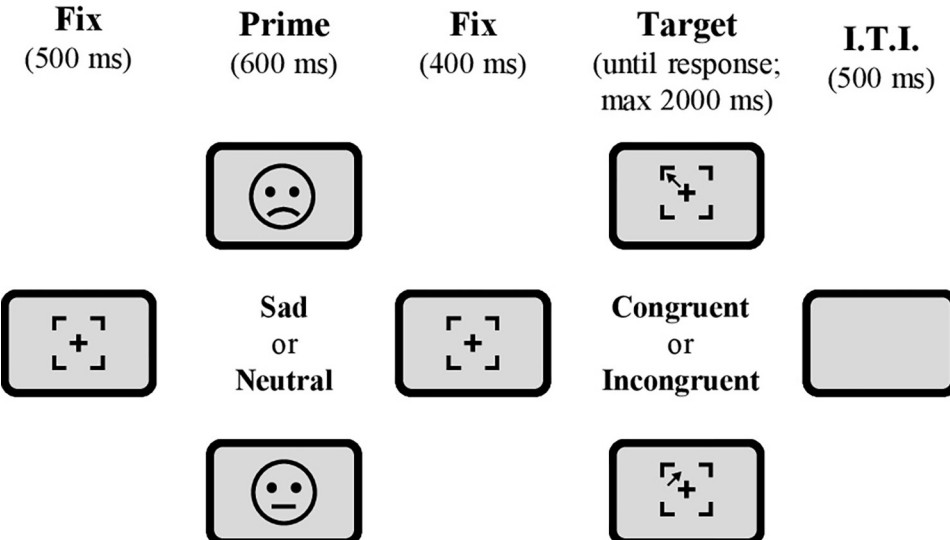

**Fig 1. Exemplary illustration of an experimental trial.** The target stimulus consisted of an arrow appearing at one of four possible positions with respect to the fixation cross (upper-left, upper-right, lower-right, and lower-left) and pointing towards one of the same four directions. On each trial, participants had to respond to the direction of the arrow regardless of the position where it appeared. Hence, trials could be either congruent or incongruent, depending on whether the arrow direction (i.e., the task-relevant information) matched or not its position (i.e., the task-irrelevant information). Participants indicated the arrow direction by pressing four keys on their computer keyboard (E, O, K, and D, respectively), which were spatially arranged to ensure the dimensional overlap between the characteristics of the stimulus and the response. The appearance of the target arrow was preceded by the presentation of either a sad or a neutral facial expression, thus providing two emotional conditions.

possible directions (upper-left, upper-right, lower-right, and lower-left). Participants were instructed to indicate the direction of the target arrow regardless of its position within the fixation stimulus. Trials could thus be either congruent or incongruent, depending on whether the arrow direction (the task-relevant information) matched or not matched its position (the task-irrelevant information). Participants provided their responses by using four keys on a computer keyboard, which were spatially arranged to ensure the dimensional overlap between the characteristics of the stimulus and the response. Specifically, the keys E, O, K, and D were associated, in a spatially compatible fashion, with the upper-left, upper-right, lower-right, and lower-left direction, respectively, and had to be pressed using the left middle, right middle, right index, and left index fingers, respectively. The target stimulus remained on screen until participant's response or up to a response time-out of 2000 ms. Afterwards, a blank screen constituting the inter-trial interval was presented for 500 ms.

LWPC and ISPC were simultaneously manipulated to measure both proactive and reactive control, respectively [29]. The assumption is that, during low LWPC blocks, the higher probability of incurring in incongruent trials favors the implementation of an anticipatory form of control and, consequently, proactive control level is higher (and vice versa). On the other hand, ISPC relies on the item presentation since the target location cannot be known in advance (each target location was equally probable in our task). Therefore, after the presentation of a target at a location with low ISPC, the higher probability of incurring in an incongruent trial at that specific location increases the amount of reactive control. We measured participants' performance while both LWPC and ISPC were varied simultaneously within the same block and then control for it at the statistical level (see Statistical analysis), as it was the most effective way to investigate the specificity of the two induced mechanisms. Moreover, this approach is also less time-consuming and more practical than traditional ones (e.g., see [41]), which, employ inducer and diagnostic items but then measure PC-related effects exclusively on diagnostic items.

The design of the trial list proceeded in two steps. First, we divided the task into six blocks as illustrated in Fig 2. List-wide probabilities were the same between blocks 1 and 5, 2 and 4, and 3 and 6, resulting in a balanced presentation order of low and high LWPC blocks. Since the effect of PC manipulations, especially of ISPC, has been challenged by associative theoretical frameworks [41–43], the number of trials for each combination of target direction and location was determined in such a way to orthogonalize the block-wise LWPC and ISPC with respect to the probability of response given a stimulus location (PRS; also known as contingency). Moreover, we tried to lower the correlation between block-wise LWPC and ISPC as much as possible by carefully varying the occurrences of the different condition combinations and using ISPC values as different as possible compared to the LWPC values of each block.

The trial order within each block was pseudorandomized using the software Mix [44] so that there were at most five consecutive repetitions of congruency and no repetitions of stimulus characteristics and/or responses, thus avoiding first order priming effects. A total of 1000 trial lists were generated. In a second step, trial-wise LWPC, ISPC and PRS were computed for each trial list. Indeed, participants were not informed about the probabilistic structure of the task, and it is not plausible to assume that the PC at the first trials of a block correspond to the overall block PC. The trial-wise LWPC and ISPC and PRS were computed using the Hierarchical Gaussian Filter [45], a filter that uses a variational Bayes under a mean-field approximation to update the probability of an event (here, the probability of target congruency for trial-wise LPWC, the probability of target congruency at each specific location for trial-wise ISPC, and the probability of target direction at each specific location for trial-wise PRS) on each trial (see Fig 3, for a graphical representation of the trial-wise LWPC). For a detailed description of the HGF, we recommend referring to the original publication [45]. However, in this context, it is

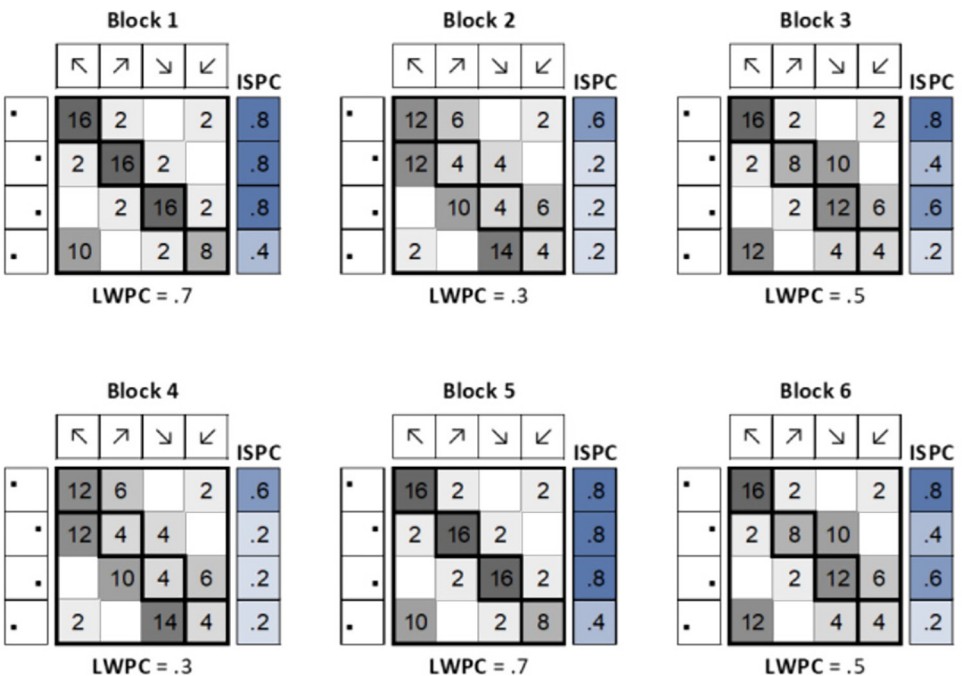

**Fig 2. Block-wise structure of the task.** Separately for each block, the image shows the number of trials with a specific target direction and location (e.g., in Block 1, we had ten trials with the arrow appearing in the lower-left corner, but pointing towards the upper-left corner; the number of trials in the diagonal are the congruent ones). For each block, the proportion of congruent trials (LWPC) and the proportion of congruent trials specific for each location (ISPC) are also indicated (ISPC is further expressed using a blue color scale). The grey color scale indicates the probability of the response given the location (PRS; also termed contingency).

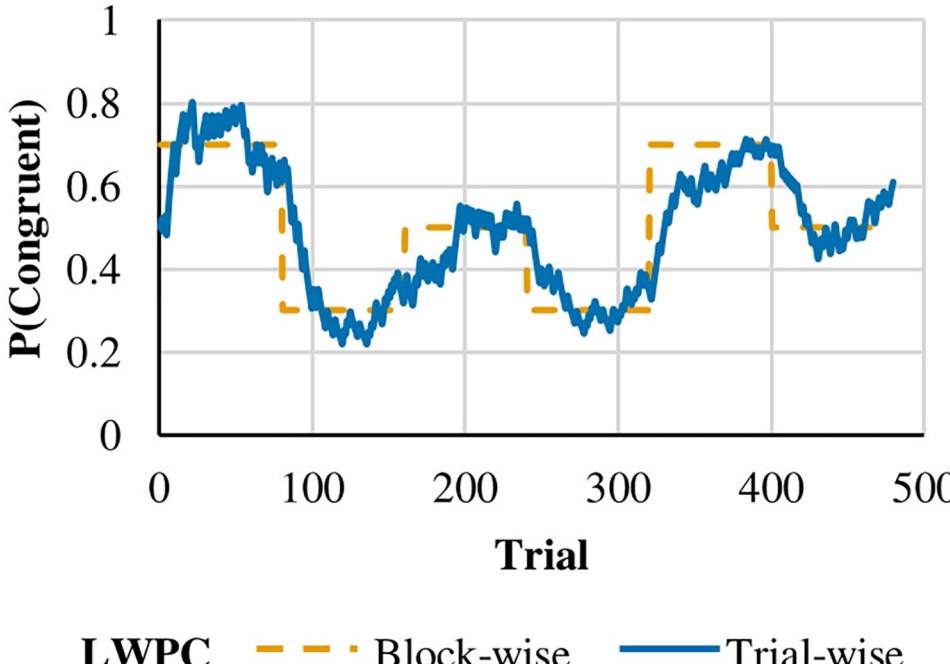

**Fig 3. Representation of the list-wide proportion of congruency (LWPC).** The figure compares the block-wise LWPC (orange line) and trial-wise LWPC (blue line) returned by the Hierarchical Gaussian Filter [45].

sufficient to state that the HGF employs Markovian update equations that offer a Bayesian equivalent of classical Rescorla-Wagner learning [46], in which beliefs after a new observation are updated according to prediction errors weighted by a learning rate. It is important to note that the HGF incorporates processes that are fundamental to current theories of learning under uncertainty and environmental volatility [47, 48], such as dynamic learning rate and precision-weighted prediction errors. Hence, this model is well-suited for representing learning in the current task, where the probability of an event (e.g., congruency) changes over time. The Hierarchical Gaussian Filter was also used to compute trial-wise probabilities of other variables used as confounding predictors in the statistical analyses (see below). Finally, the trial list with the lowest correlation between ISPC and PRS was selected and used in the present study. The shared variance between trial-wise ISPC and PRS was 1%, it was 0.2% between LWPC and PRS, and it was 22.7% between LWPC and ISPC.

Before the beginning of the task, general instructions were provided. Particular care was taken to keep the instructions as simple and clear as possible (also with the support of illustrations). Participants were also recommended to perform the experiment in a quiet environment without distractions and to maintain a comfortable posture that allowed them to look straight to the center of the screen and keep the responding fingers in contact with the response keys. Instructions were followed by a block of practice trials with LWPC and ISPC equal to .5, during which participants received feedback on their responses. Practice trials were presented until participants reached an accuracy of 75% within sixteen trials. A one-minute break was provided in the middle of the task. At the end of the task, we performed a manipulation check by asking participants to rate the intensity with which the presented faces induced each of the following emotions: anger, happiness, fear, sadness, disgust, surprise. Sadness significantly received higher rating compared to the other emotions (all $t$ values > 2.75, all $p$ values < .013), confirming that our experimental manipulation was indeed effective in specifically evoking sadness in our participants".

## Statistical analyses

Analyses of RTs were performed by means of linear mixed-effects models (LMM) using the *lme4* library [49] in R (http://www.R-project.org/). LMM is the most appropriate method to analyzed designs with by-subject and by-items (face stimuli) crossed random effects [50]. Data and code are available at https://osf.io/bc9x3/. Data from the first trial of the task and the first trial after the break, as well as data from error trials (i.e., incorrect or missing responses to the target) and post-error trials were not included (mean percentage of excluded trials: 6.3% trials, SD = 4.8% trials). Moreover, to control for the impact of positive skewness in the distribution of RTs (in ms), all the analyses were performed on the inverse-transformed RTs (iRT), computed as -1000/RT [51]. First, we specified a full LMM including all the experimental effects along with several possible confounding predictors that were expected to explain trial-by-trial variability in iRTs. Specifically, the fixed part of the model included the following experimental effects of interest: congruency (two-level factor with effect-coding: congruent = -1, incongruent = 1), the continuous predictors trial-wise LWPC and ISPC, and prime Emotion (two-level factor: neutral = -1, sadness = 1), as well as the three level interactions between Congruency × LWPC × Emotion and Congruency × ISPC × Emotion (and the associated lower order interactions). The fixed part included also the following confounding predictors: following Baayen and Milin [52], the rank-order of each trial (Trial) and the iRT at the preceding trial (Preceding RT) were included to control for the temporal dependencies between successive trials (i.e., learning/fatigue effects and RT autocorrelation, respectively); the horizontal and vertical coding of the arrow direction (respectively, hDIR and vDIR) were included to account for potential differences due to the response hand and finger, respectively; the horizontal

and vertical coding of the arrow location (hPOS and vPOS, respectively) were included to account for potential differences due to left/right and upper/lower visual field, respectively; contingency (PRS), probability of the response (PR) and probability of the target location (PL) were included as low-level confounding probabilities. The random part included crossed random effects for participants and prime images. Specifically, the model included by-image random intercepts, and by-participant correlated random intercepts and slopes for the Congruency × LWPC × Emotion, Congruency × ISPC × Emotion interactions and the associated lower order interactions and main effects. Effect coding was applied to all two-level factors. All probabilities were expressed in a logit scale. All other continuous predictors were centered to have mean 0 and scaled to have SD = 1 in order to facilitate model convergence. A model selection procedure from the full model was conducted through the function *step* of the lmerTest R-library [53], which performs backwards step-wise elimination of non-significant random and fixed effects of LMM [54]. This procedure was performed just to evaluate the inclusion of the confounding variables in the fixed part, and the adequacy of the random effects structure.

## Results

The overall accuracy was very high and at ceiling for congruent trials (mean = .99). Consequently, the participants' Stroop effects on the accuracy heavily depended on their average accuracy (i.e., participants with a very high overall accuracy cannot show a Stroop effect). This severely limits the interpretability of the analyses on accuracy and introduces strong biases in the estimation of the reliability of this measure. For this reason, we did not analyze accuracy.

Backward LMM selection did not remove any confounding predictors, but it removed all random effects associated with Emotion (log-likelihood ratio test between the full and final models: $\chi 2(57) = 21.7$; p > .999). Therefore, the final model was specified as the following Wilkinson-notation formula:

iRT ~ Trial + Preceding RT + PRS + PR + PL + hPOS + vPOS + hDIR + vDIR + Congruency × LWPC × Emotion + Congruency × ISPC × Emotion + (Congruency × LWPC + Congruency × ISPC | Participant) + (1 | Image)

Visual inspection of the residuals showed that they were skewed. As suggested by Baayen and Milin [52], trials with absolute standardized residuals higher than 2.5 SD were considered outliers and removed (2.1% of the trials). After outlier trials removal, the model was refitted achieving reasonable closeness to normality.

A summary of the LMM results is presented in Table 1. Concerning the proactive control manipulation, we observed a significant Congruency × LWPC interaction characterized by an increase in iRT with increasing LWPC (Fig 4, right plot). This interaction was further modulated by priming Emotion (Fig 4). As also confirmed by post-hoc contrasts–implemented using the *emtrends* function of the *emmeans* R library [55], *p* values adjusted with Tukey method for 4 comparisons–while LWPC slopes did not significantly differ between neutral and sad priming emotional expressions in incongruent trials ($z = -1.04$, $p = .725$), there was a significant LWPC slope difference between priming emotional expressions in congruent trials ($z = -4.69$, $p < .001$). We also contrasted marginal iRTs at extreme LWPC values for both Emotion and Congruency levels using the *estimate_contrasts* function of the *modelbased* R package [56]–*p* values adjusted with Tukey method for 28 comparisons. The only significant difference between priming Emotional expressions was observed at congruent trials with the highest LWPC ($z = -4.06$, $p = .001$), where iRT were longer after sad priming.

Concerning the reactive control manipulation, there was neither a significant interaction between ISPC and Congruency, nor a significant modulation of that interaction by Emotion.

**Table 1. Summary output of the final LMM model.**

| Predictors | Estimates | CI | p |
|---|---|---|---|
| (Intercept) | -2.329 | -2.420 – -2.237 | **<0.001** |
| Trial | -0.103 | -0.107 – -0.099 | **<0.001** |
| Preceding RT | 0.086 | 0.082 – 0.090 | **<0.001** |
| PRS | -0.05 | -0.054 – -0.045 | **<0.001** |
| PR | -0.102 | -0.118 – -0.085 | **<0.001** |
| PL | -0.138 | -0.183 – -0.094 | **<0.001** |
| hPOS | -0.013 | -0.017 – -0.008 | **<0.001** |
| vPOS | 0.021 | 0.016 – 0.026 | **<0.001** |
| hDIR | -0.006 | -0.011 – -0.002 | **0.008** |
| vDIR | -0.074 | -0.079 – -0.069 | **<0.001** |
| CON | 0.225 | 0.211 – 0.240 | **<0.001** |
| LWPC | 0.058 | 0.044 – 0.072 | **<0.001** |
| Emotion | <0.001 | -0.010 – 0.009 | 0.988 |
| ISPC | -0.01 | -0.018 – -0.002 | **0.015** |
| Congruency × LWPC | 0.054 | 0.041 – 0.067 | **<0.001** |
| Congruency × Emotion | -0.003 | -0.007 – 0.001 | 0.138 |
| LWPC × EMOTION | 0.013 | 0.007 – 0.019 | **<0.001** |
| Congruency × ISPC | -0.005 | -0.019 – 0.009 | 0.5 |
| Emotion × ISPC | -0.007 | -0.012 – -0.002 | **0.004** |
| (Congruency × LWPC) × Emotion | -0.008 | -0.014 – -0.002 | **0.014** |
| (Congruency × Emotion) × ISPC | 0.001 | -0.004 – 0.006 | 0.74 |
| Marginal $R^2$ / Conditional $R^2$ | 0.289 / 0.673 | | |

## Discussion

The present study investigated whether processing of negatively valenced (i.e., sad) priming influences conflict resolution in a Stroop task. A special emphasis was placed on the distinction between proactive and reactive modes of cognitive control, which were manipulated by varying the proportion of congruency in a block-wide (LWPC) and item-specific (ISPC) manner, respectively. Several different hypotheses were put forward on the basis of previous mixed findings. Our results showed that sad emotional priming was associated with behavioral costs only in trials with low cognitive control exertion, that is, in congruent trials (where there is no need of reactive control) with lower proactive control.

Before elaborating on the interaction between emotional processing and cognitive control, we first briefly discuss the results of the employed proactive and reactive control manipulations, which are partially in contrast with previous findings. Specifically, in line with previous studies [28, 29, 41, 57], we found that LWPC manipulation significantly modulated Stroop conflict resolution as, when LWPC was lower, the higher level of proactive control reduced Stroop effect magnitude. On the other hand, we did not find a significant effect of ISPC manipulation. In contrast with that, previous studies have found such an effect both in different [28] and in the same participants [29]. However, unlike these studies, in which LWPC and ISCP manipulations were implemented independently in different blocks, we simultaneously manipulated both of them. Therefore, our approach might be a more effective way to directly test the independence of proactive and reactive control. However, although we did not find the specific behavioral signatures of reactive control, we cannot firmly conclude that such control mode does not exist, because we also manipulated and included in our model the confounding variable of contingency to control for the potential role of associative learning mechanism

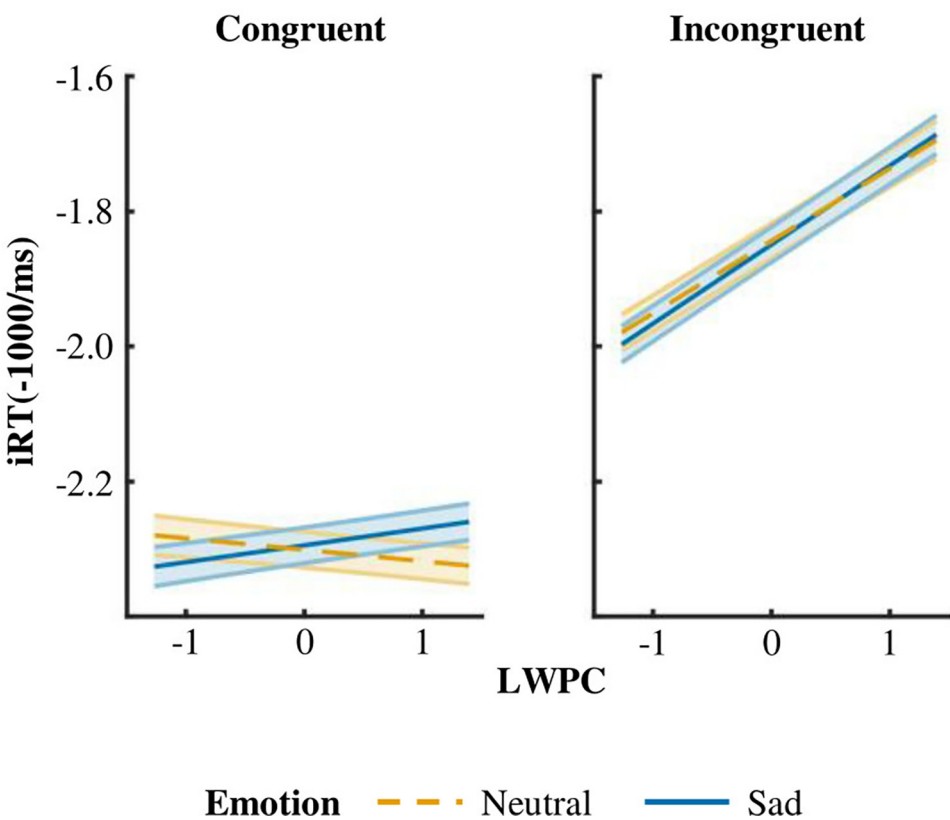

**Fig 4. Interaction effect between Congruency, LWPC, and priming Emotion.** The figure shows the conditional effect of trial-wise LWPC (expressed on a logit scale) on iRT for Neutral (orange lines) and Sad (blue line) priming conditions, separately for Congruent and Incongruent conditions. Shaded error bars indicate standard errors of estimated marginal means.

[41–43, 58]. Therefore, for now, our results indicate that contingency explains more variance than ISPC, but further studies are required to better disentangle contingency from ISPC or, alternatively, to find a better manipulation for more effectively investigating the independence of the reactive control mechanism.

Turning back to our main questions, we observed a slowing of RTs after emotionally negative priming stimuli, but only in congruent (i.e., no conflict) trials when proactive control demands decreased (i.e. higher LWPC). A first implication of this finding is that we did not observe any facilitatory effect of sad priming stimuli. Hence, our results did not corroborate the hypothesis that sad stimuli increase control by narrowing the scope of attention [31]. A possible explanation for this discrepancy can be attributed to the type of emotion used as the comparison stimulus. Indeed, the terms facilitation/interference are always relative to comparison terms. For example, Melcher and colleagues [31] found that sad priming led to a Stroop effect reduction compared to both neutral and fear priming (the last two did not significantly differ). However, their neutral stimuli were 25% happy faces to avoid the observed risk that 100% neutral faces might be elaborated as emotionally negative [59]. Some studies have shown that happy faces increase congruency effects (e.g., Stroop effect) similarly to fearful or angry faces and to a larger extent than sad faces [60–62]. It follows that what is seen as facilitation might be also interpreted as a smaller interference compared to the adopted baseline condition. Moreover, if the facilitatory effect of sad stimuli on conflict resolution has been commonly attributed to the narrowing of attention, it should be also noted that the effect of

sadness on the scope of attention is inconsistent across studies. Indeed, it has been reported that sadness can extend, narrow, or even have no significant effect on the attention scope (for an overview, see [63]).

We did not observe significant evidence in favor of the hypothesis that negative emotional priming interferes with proactive and reactive control processes involved in conflict resolution, either. In the present study, indeed, the only significant interference effect elicited by sad priming was observed in those situations in which cognitive control requirements were at the minimum. Thus, our results seem to be consistent with the idea that proactive control is effective in preventing negative emotional priming from engaging cognitive resources needed for conflict resolution [7, 9]. Moreover, the fact that differences between sad and neutral priming at lower levels of proactive control were observed only in congruent trials might suggest that reactive control could also overcome emotional interference. Indeed, at lower levels of proactive control, the conflict exerted by incongruent trials must necessarily be resolved by reactive control processes to respond correctly, as assumed by the DMC proposal [64]. It is important to note that here we are referring to reactive control processes that are assumed to be different from those dependent on the ISPC values, which operate as a (faster) "stimulus-attention association" triggered by the item ISPC as soon as it is identified (e.g., [30, 65, 66]; see also [27]). By contrast, it can be assumed that (later) reactive control processes must intervene to resolve the conflict caused by incongruent trials when other control processes failed, that is, when proactive control and the faster stimulus-driven reactive control are both low. In this case, indeed, incongruent trials still elicit unexpected conflict that has to be resolved by cognitive control mechanisms in order to respond correctly to them.

Of note, this putative ability of reactive control in overcoming emotional processing can be appreciated in emotional priming Stroop tasks, but not in face-word Stroop-like tasks. Indeed, when emotions are task-relevant, there is no reason to prevent/interrupt emotional processing. In consideration of the paucity of studies that have employed emotional priming Stroop tasks, more studies with the emotional priming Stroop tasks are needed to further understand this interaction.

Overall, taking our results into account, it seems that emotional interference affects cognitive processes when the level of cognitive control is negligible. This conclusion is in line with previous research questioning the effect of emotional stimuli as a Stroop effect. After conducting a series of experiments, Algom and colleagues [67] concluded that the effect of negative emotions causes a general cognitive slowing of processes not selectively related to attentional mechanisms involved in the Stroop task (see also [68, 69]).

Some caveats to our results along with some study limitations need to be acknowledged. First, our conclusions are limited to sadness. Thus, further studies are needed to generalize them to emotions with opposite valence or higher arousal. Second, we used a fixed interval between prime and target stimuli whose length was selected to be adequate for an electrophysiological study we planned to conduct as follow-up. Since the length of the interstimulus interval might be determinant for observing emotional interference [70], we cannot rule out that our finding might differ with shorter intervals. Electrophysiological measures will help to describe the time course of emotional interference as well as its interaction with proactive and reactive control mechanisms. Third, our findings are limited to healthy individuals. Since interindividual differences in personality traits (e.g., anxiety; [71, 72]) or psychopathological conditions (e.g., depression; [72–74] are known to correlate with emotional Stroop effect, it would be interesting to test whether emotional processing is able to compete with cognitive control processes in clinical populations. Finally, this study could also be extended to cross-channel and cross-modal settings. Future studies can further explore the potential individual

differences (e.g., gender differences) and task-related differences in auditory or audiovisual emotional Stroop priming effects [75–77].

In conclusion, the present findings show that emotional elaboration of sad stimuli do not interfere with cognitive control processes in the Stroop task. Conversely, proactive and reactive cognitive control modes appear effective in overcoming emotional interference of priming stimuli. Future electrophysiological and clinical studies with our approach might help to further characterize the relationship between emotional processing and cognitive control and their alterations in psychopathological conditions.

## Supporting information

**S1 Checklist. STROBE statement—checklist of items that should be included in reports of observational studies.**
(DOCX)

**S1 Appendix. List of images used from the NimStim set of facial expressions.**
(PDF)

**S1 File.**
(PDF)

**S2 File.**
(PDF)

## Author Contributions

**Conceptualization:** Antonino Visalli, Ettore Ambrosini, Fabio Sambataro, Elena Tenconi, Antonino Vallesi.

**Data curation:** Antonino Visalli, Ettore Ambrosini.

**Formal analysis:** Antonino Visalli.

**Funding acquisition:** Elena Tenconi, Antonino Vallesi.

**Investigation:** Antonino Visalli, Ettore Ambrosini, Giada Viviani.

**Methodology:** Antonino Visalli, Fabio Sambataro, Antonino Vallesi.

**Project administration:** Antonino Vallesi.

**Resources:** Antonino Vallesi.

**Software:** Antonino Visalli.

**Supervision:** Ettore Ambrosini, Antonino Vallesi.

**Writing – original draft:** Antonino Visalli.

**Writing – review & editing:** Antonino Visalli, Ettore Ambrosini, Giada Viviani, Fabio Sambataro, Elena Tenconi, Antonino Vallesi.

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
