## [Decision Letter · Decision Letter 0]

7 Jul 2023

PONE-D-23-12904On the relationship between emotions and cognitive control: Evidence from an observational study on emotional priming Stroop taskPLOS ONE

Dear Dr. Vallesi,

Thank you for submitting your manuscript to PLOS ONE. After careful consideration, we feel that it has merit but does not fully meet PLOS ONE’s publication criteria as it currently stands. Therefore, we invite you to submit a revised version of the manuscript that addresses the points raised during the review process.

We look forward to receiving your revised manuscript.

Kind regards,

Valerio Manippa

Academic Editor

PLOS ONE

Journal Requirements:

2. We note that Figure 1 includes an image of a participant in the study. 

Reviewers' comments:

Reviewer's Responses to Questions

**Comments to the Author**

1. Is the manuscript technically sound, and do the data support the conclusions?

Reviewer #1: Yes

Reviewer #2: Partly

2. Has the statistical analysis been performed appropriately and rigorously? 

Reviewer #1: Yes

Reviewer #2: Yes

3. Have the authors made all data underlying the findings in their manuscript fully available?

Reviewer #1: No

Reviewer #2: Yes

4. Is the manuscript presented in an intelligible fashion and written in standard English?

Reviewer #1: Yes

Reviewer #2: No

5. Review Comments to the Author

Reviewer #1: Review of PONE-D-23-12904 “On the relationship between emotions and cognitive control: Evidence from an observational study on emotional priming Stroop task.”

General Evaluation

The authors investigate the interplay between emotional processing and cognitive control, particularly in regard to the question of whether proactive and reactive control modalities can shield control processes from the negative effects of processing sad faces. They find that negative emotions impact performance only on those trials where control demands were low (i.e., congruent trials), when proactive control was not exerted (i.e., when most trials were congruent). The authors conclude that emotional processing does not affect control processes specifically but that, nonetheless, proactive and reactive control help in attenuating the negative effects of emotional processing on performance. There is much to praise in the manuscript. Most of all, I appreciated the well-conducted statistical analyses, especially the conceptualization of proportion congruence as a continuous variable which depends on trial history, and the inclusion of many possible confounds in the models. Furthermore, the results were discussed openly, recognising possible limitations in the presented work, and avoiding over-generalizations. Nonetheless, a few concerns remain. First of all, I’m wondering why ISPC and LWPC were manipulated in the same blocks. If the aim is to disentangle the effects of emotional processing on proactive and reactive control, I would have thought that having distinct blocks to assess such effects would be a cleaner way to go. Second, I would appreciate if some methodological details were made more explicit. Finally, I suggest some additional analyses that I believe could strengthen the results.

Major comments

1. ISPC and LWPC are manipulated simultaneously to investigate their effects on emotional processing. I would appreciate if the authors could clarify the advantages of this choice over manipulating ISPC and LWPC in different blocks (perhaps using different items so to avoid using biased items in LWPC blocks). I do understand that this may be advantageous for power reasons, but I believe it’s also a risky choice in terms of interpretability of the results. For example, the claim that reactive control shields performance against emotional processing is currently based on the finding that emotional processing has no effect on incongruent trials when proactive control is low. Even though I understand the reasoning that when proactive control is low, reactive control will be used, I believe that this is only indirect evidence for this claim, and that more direct evidence would be collected in a block in which reactive control use could be inferred exclusively on the base of the ISPC manipulation.

2. Related to this, the use of reactive control seems to be inferred in two ways. First, reactive control is supposed to be present when items are mostly incongruent. Second, reactive control is supposed to be present when proactive control is low. Can the authors elaborate on this point?

3. I didn’t understand how the LWPC and ISPC were made orthogonal (line 238). It would be important to elaborate more on that.

4. The use of the inverse transformation for RTs has been criticized for biasing the interaction terms toward non-significance (Balota et al., 2013). In addition, when avoiding the use of such transformation, the influence of previous trial RTs on the LWPC effect was found to be reduced (Cohen-Shikora et al., 2019). In order to circumvent this problem, while also dealing with the skewness issues, the authors used generalized linear mixed models (GLMM). I was wondering whether it would be possible to re-run the analyses fitting GLMM to untransformed RTs. As I am not an expert with such models however, I may be overlooking some potential issues: in this case, please just let me know.

5. Figure 3 shows the trial-by-trial value of LWPC. As I said, it makes a lot of sense to me to compute LWPC as a continuous variable. However, I keep wondering why the trial-by-trial LWPC doesn’t converge with the overall LWPC at the end of the first block, and also at the end of the experiment (I thought it would be 0.5 in total, but I see the trial-wise value is over 0.6). I guess I’m thinking of it as cumulative probability, which is probably not. Is this something related to the Bayesian filter employed? Maybe other readers would be wondering about the same issue, so perhaps it’s worth making the procedure for calculating trial-wise LWPC more explicit in the text. Maybe a footnote would also suffice.

Minor comments

1. In some parts of the manuscript, it’s said that emotional stimuli cause interference/conflict. As in the cognitive control literature these terms usually indicate the simultaneous activation of more cognitive representations, I suggest that the authors could find another word for indicating the kind of bottleneck problem posed by emotional stimuli.

2. In some parts of the manuscript, cognitive control is regarded as a processing stage. I would argue that control mechanisms are processes that impact processing stages rather than being a processing stage themselves.

3. Line 337: although noticeable from the figure, it would be useful to explicitly report the direction of the difference between priming emotional expressions in congruent trials.

4. Can the authors also report the direction for the Emotion x ISPC interaction? Is this relevant for their claims about reactive control?

5. If there are no impediments to this, could the authors please share the raw data of the experiment? I could not find them on OSF.

Formatting and language

Line 22: Relate TO

Line 64: To THIS aim

Line 84-85: I think the authors mean “in the EMOTIONAL color-word Stroop task”. Also, I would probably call it emotional word-color Stroop, as it is the word to be emotional (not the color).

Line 95-96: “… to investigate whether the processing of emotions interacts with cognitive control and AT WHICH LEVEL”. This sentence may suggest that the authors intend to investigate the impact of emotional processing on control operating on different cognitive representations (e.g., conflict at the response level, or at the task level). I think “mode” would be avoid confusion here.

Line 119: Perhaps say “EMOTIONAL neutral” stimuli to stress that we’re not talking about neutral Stroop stimuli.

Line 134: Instead of “steal” perhaps use another word? Withhold? Employ? Occupy?

Line 146: Maybe without “therefore”?

Line 157: Maybe “Ethics” instead of “Ethical”?

Line 184: stroop with capital S.

Line 274: “At the end of the we performed”. Missing “practice phase”?

Line 319: DID not analyze (without d)

Line 393: “A second implication of our findings … “ is followed by the description of the findings themselves. I would remove the implication part.

References

Cohen-Shikora, E. R., Suh, J., & Bugg, J. M. (2019). Assessing the temporal learning account of the list-wide proportion congruence effect. Journal of Experimental Psychology: Learning, Memory, and Cognition, 45(9), 1703.

Reviewer #2: Thank you for providing me with the opportunity to review this submission.

The authors performed an emotional priming Stroop test with list-wide (LWPC) and item-specific (ISPC) proportion congruence simultaneously manipulated. Findings suggest that sad priming led to behavioral costs only in trials with low proactive and reactive cognitive control. Proactive and reactive cognitive control can overcome the emotional interference of priming stimuli.

Overall, the work is of particular interest and bring important new insight in how emotion perception and cognitive control interact to shape behavior. The study is well-designed and the manuscript is generally well written with some issues presented below to be resolved:

Line 28: In the abstract, low proactive and reactive cognitive control was mentioned when describing the results (but not methods) of the study. Consider adding some details in the sentences on methods to indicate how cognitive control was manipulated.

Line 78-80: The parenthesis “for our aim” can be removed in the sentence. This is same for other parentheses like “,then” (line 241), and “, more specifically” (line 65).

Line 94: The sentence should read “we employed an emotional priming spatial Stroop task to investigate ….” (the comma can be deleted).

Line 121: Please define what is directional hypothesis since some readers may not be familiar with it.

Line 194-200: The material list can be moved to appendix so as not to distract the reader.

Line 237-240: I’m a bit confused by seeing figure 2. How are blocks 1 and 5 (Also block 2 vs. block 4, block 3 vs. block 6) different? How are LWPC and ISPC orthogonalized?

Line 249: What is the difference between the first and second “target congruency”. Please give specifications.

P275: What was the purpose of the manipulation check to compare sadness with other emotions without neutrality? Please make justifications.

Line 277: Please specify the p values apart from t values.

Line 314: When reporting the results of linear mixed-effects models, consider providing the F values/χ2 for model comparisons as a justification for the final model (at least in supplemental materials).

Line 410: It should be cautious to draw a conclusion that emotion processing affects processing stages other than cognitive control since other processing stages were not measured or manipulated in the current study.

Line 415: The sentences should read “First, our conclusions are limited to sadness. Thus, further studies are needed to generalize them to emotions with opposite valence or higher arousal.”

Line 414-431: This study can also be extended to cross-channel and cross-modal settings. Future studies can further explore the potential individual differences (e.g., gender differences) and task-related differences in auditory or audiovisual emotional Stroop priming effects. See references below:

1) Filippi, P., Ocklenburg, S., Bowling, D. L., Heege, L., Güntürkün, O., Newen, A., & de Boer, B. (2017). More than words (and faces): evidence for a Stroop effect of prosody in emotion word processing. Cognition and Emotion, 31(5), 879-891.

2) Lin, Y., Ding, H., & Zhang, Y. (2021a). Gender differences in identifying facial, prosodic, and semantic emotions show category- and channel-specific effects mediated by encoder's gender. Journal of Speech, Language & Hearing Research, 64(8), 2941-2955.

3) Lin, Y., Ding, H., & Zhang, Y. (2021b). Unisensory and multisensory Stroop effects modulate gender differences in verbal and nonverbal emotion perception. Journal of Speech, Language, and Hearing Research, 64(11), 4439-4457.

6. PLOS authors have the option to publish the peer review history of their article (what does this mean?). If published, this will include your full peer review and any attached files.

Reviewer #1: **Yes: **Luca Moretti

Reviewer #2: No

---

## [Author Response · Author response to Decision Letter 0]

18 Oct 2023

Dear Dott. Valerio Manippa,

Thank you for considering our revised manuscript for publication in Plos ONE. We would like to express our gratitude to the reviewers for their valuable comments, all of which we have addressed to the best of our ability.

In particular, in response to Reviewer 1, we have extended and clarified the reasons and advantages behind our experimental manipulation of LWPC and ISPC. All his/her major and minor issues were also addressed. We believe these comments were very useful to improve the description of some aspects of our work that, in the first version, were partly taken for granted and not explained in sufficient detail. 

Moreover, following the comments raised by Reviewer 2, we also amended the manuscript to improve its clarity and readability.

We believe that our manuscript has been substantially improved after considering the issues raised by both reviewers. We, thus, truly wish that the reviewers and you will consider our manuscript suitable to appear in your journal.

Regarding the Journal requirement 2 “We note that Figure 1 includes an image of a participant in the study.”, we would like to clarify that Figure 1 does not include an image of a participant. Instead, it displays an exemplar of the provided stimuli used in the study. 

What follows are answers to each point the Reviewers raised. 

 

Reviewer #1: 

General Evaluation

The authors investigate the interplay between emotional processing and cognitive control, particularly in regard to the question of whether proactive and reactive control modalities can shield control processes from the negative effects of processing sad faces. They find that negative emotions impact performance only on those trials where control demands were low (i.e., congruent trials), when proactive control was not exerted (i.e., when most trials were congruent). The authors conclude that emotional processing does not affect control processes specifically but that, nonetheless, proactive and reactive control help in attenuating the negative effects of emotional processing on performance. There is much to praise in the manuscript. Most of all, I appreciated the well-conducted statistical analyses, especially the conceptualization of proportion congruence as a continuous variable which depends on trial history, and the inclusion of many possible confounds in the models. Furthermore, the results were discussed openly, recognising possible limitations in the presented work, and avoiding over-generalizations. Nonetheless, a few concerns remain. First of all, I’m wondering why ISPC and LWPC were manipulated in the same blocks. If the aim is to disentangle the effects of emotional processing on proactive and reactive control, I would have thought that having distinct blocks to assess such effects would be a cleaner way to go. Second, I would appreciate if some methodological details were made more explicit. Finally, I suggest some additional analyses that I believe could strengthen the results.

R: We thank the reviewer for the positive assessment of our work and the very constructive comments.

Major comments

1. ISPC and LWPC are manipulated simultaneously to investigate their effects on emotional processing. I would appreciate if the authors could clarify the advantages of this choice over manipulating ISPC and LWPC in different blocks (perhaps using different items so to avoid using biased items in LWPC blocks). I do understand that this may be advantageous for power reasons, but I believe it’s also a risky choice in terms of interpretability of the results. For example, the claim that reactive control shields performance against emotional processing is currently based on the finding that emotional processing has no effect on incongruent trials when proactive control is low. Even though I understand the reasoning that when proactive control is low, reactive control will be used, I believe that this is only indirect evidence for this claim, and that more direct evidence would be collected in a block in which reactive control use could be inferred exclusively on the base of the ISPC manipulation.

R: We thank the reviewer for his comment. Indeed, as an innovative approach, it may appear risky. However, our choice is based on the fact that manipulating LWPC and ISPC in different blocks does not allow examining the specificity of the control mechanisms induced by these two manipulations. The most plausible way to investigate their specificity is instead to measure participants' performance while both LWPC and ISPC are parametrically varied simultaneously in the same block. Then, by employing an appropriate statistical approach, such as the one used here (LMM), it is possible to investigate the effect of both manipulations even when tested simultaneously, thus also controlling for factors such as general attention and arousal within each block. We have also chosen not to use traditional manipulations, such as the one suggested by the reviewer (inducer vs diagnostic items), as there are drawbacks associated with their implementation. Specifically, differentiating between inducer and diagnostic items for LWPC measures proves to be impractical and time-consuming, as it requires measuring PC-related effects exclusively on diagnostic items while excluding inducer items from the analysis. We added this clarification at lines 230-235.

As regards the second part of the reviewer’s comment, please refer to our reply to the next point.

2. Related to this, the use of reactive control seems to be inferred in two ways. First, reactive control is supposed to be present when items are mostly incongruent. Second, reactive control is supposed to be present when proactive control is low. Can the authors elaborate on this point?

R: We thank the reviewer for this comment as well, which gave us the opportunity to clarify this point. Indeed, we assumed that reactive control is engaged when specific items are mostly incongruent, that is, with low ISPC, in line with the literature on reactive control effects on Stroop performance. We also assumed that reactive control is preferentially engaged when proactive control is not possible or advantageous, that is, when LWPC is high. This assumption is based on the DMC proposal (De Pisapia & Braver, 2006). However, it should be noted that, while the first type of reactive control is dependent on the ISPC level because it operates as a (faster) “stimulus-attention association” triggered by the item ISPC as soon as it is identified (e.g., Bugg, 2012, 2017; Bugg & Hutchison, 2013; see also Tafuro et al., 2020), incongruent trials still trigger a later form of reactive control to overcome the conflict, even when both LWPC and ISPC are high (and, thus, when both proactive control and the ISPC-related reactive control are low). We added this clarification at lines 419-429. 

3. I didn’t understand how the LWPC and ISPC were made orthogonal (line 238). It would be important to elaborate more on that.

R: We thank the reviewer for the careful reading of the manuscript. In the revised manuscript we clarified this point, which now reads as follows (lines 238-245): “Since the effect of PC manipulations, especially of ISPC, has been challenged by associative theoretical frameworks (41–43), the number of trials for each combination of target direction and location was determined in such a way to orthogonalize the block-wise LWPC and ISPC with respect to probability of response given a stimulus location (PRS; also known as contingency). Moreover, we tried to lower the correlation between block-wise LWPC and ISPC as much as possible by carefully varying the occurrences of the different condition combinations and using ISPC values as different as possible compared to the LWPC values of each block.”

In addition, we added the shared variance between trial-wise LWPC-ISPC, LWPC-PRS, and ISPC-PRS (lines 276-277).

4. The use of the inverse transformation for RTs has been criticized for biasing the interaction terms toward non-significance (Balota et al., 2013). In addition, when avoiding the use of such transformation, the influence of previous trial RTs on the LWPC effect was found to be reduced (Cohen-Shikora et al., 2019). In order to circumvent this problem, while also dealing with the skewness issues, the authors used generalized linear mixed models (GLMM). I was wondering whether it would be possible to re-run the analyses fitting GLMM to untransformed RTs. As I am not an expert with such models however, I may be overlooking some potential issues: in this case, please just let me know.

R: Gamma models pose notorious challenges in terms of fitting. We attempted to fit the model recommended by the reviewer; however, it failed to converge and yield a unique solution. We would like to bring the reviewer's attention to the fact that, in the cited article, they employed a model with solely random intercepts for participants, as the more appropriate model incorporating random slopes failed to converge (footnote 10). It is crucial to note that "random-intercepts-only LMEMs can have catastrophically high Type I error rates" (Barr et al., 2013). Therefore, solutions allowing the inclusion of random slopes should be favored. Nonetheless, to comply with the reviewer's suggestion, we also fitted the model with only random intercepts; however, it still failed to converge (possibly due to our design including random intercepts not only for participants but also for items). Considering all these factors, we are sorry that we cannot incorporate the proposed analysis into the manuscript.

5. Figure 3 shows the trial-by-trial value of LWPC. As I said, it makes a lot of sense to me to compute LWPC as a continuous variable. However, I keep wondering why the trial-by-trial LWPC doesn’t converge with the overall LWPC at the end of the first block, and also at the end of the experiment (I thought it would be 0.5 in total, but I see the trial-wise value is over 0.6). I guess I’m thinking of it as cumulative probability, which is probably not. Is this something related to the Bayesian filter employed? Maybe other readers would be wondering about the same issue, so perhaps it’s worth making the procedure for calculating trial-wise LWPC more explicit in the text. Maybe a footnote would also suffice.

R: The HGF, as other Bayesian or Reinforcement Learning algorithms, employs a sort of delta-rule to compute trial-by-trial updates of stimulus contingencies. Specifically, prior beliefs are updated in relation to the current prediction error weighted by a dynamic learning rate. These models are well-suited for capturing the learning processes of living agents in volatile environments, where stimulus contingencies can change over time. It is worth noting that an agent that treats all observations equally and retains them indefinitely (such as when computing a cumulative probability) would struggle to adapt to changes in contingencies. In the revised manuscript (lines 265-273), we have included the following paragraph: “For a detailed description of the HGF, we recommend referring to the original publication (45). However, in this context, it suffices to state that the HGF employs Markovian update equations that offer a Bayesian equivalent of classical Rescorla-Wagner learning (46), in which beliefs after a new observation are updated according to prediction errors weighted by a learning rate. It is important to note that the HGF incorporates processes that are fundamental to current theories of learning under uncertainty and environmental volatility (47,48), such as dynamic learning rate and precision-weighted prediction errors. Hence, this model is well-suited for representing learning in the current task, where the probability of an event (e.g., congruency) changes over time.”

Minor comments

1. In some parts of the manuscript, it’s said that emotional stimuli cause interference/conflict. As in the cognitive control literature these terms usually indicate the simultaneous activation of more cognitive representations, I suggest that the authors could find another word for indicating the kind of bottleneck problem posed by emotional stimuli.

R: In the revised version of the manuscript we use the term conflict to denote the Stroop conflict deriving from competing representations (and thus causing the Stroop effect; De Houwer, 2003), while the term interference is used to indicate the emotional interference, which does not involve conflict between competing representations (Algom et al., 2004)

2. In some parts of the manuscript, cognitive control is regarded as a processing stage. I would argue that control mechanisms are processes that impact processing stages rather than being a processing stage themselves.

R: We thank the reviewer for the comment. The manuscript has been amended accordingly.

3. Line 337: although noticeable from the figure, it would be useful to explicitly report the direction of the difference between priming emotional expressions in congruent trials.

R: As already reported in the original manuscript, “iRT were longer after sad priming”.

4. Can the authors also report the direction for the Emotion x ISPC interaction? Is this relevant for their claims about reactive control?

R: The ISPC effect was larger for sad priming emotional expression. However, it should be noted that, by definition, reactive control is exerted when a conflict is detected (i.e., with incongruent stimuli). Therefore, ISPC effects cannot be interpreted without its interaction with Congruency. Moreover, in our statistical models we used the effect coding for the Congruency factor (instead of the reference coding). So, the main effect of ISPC (as well as its interactions not including the Congruency factor) refers to the average between Congruent and Incongruent trials, making it not interpretable because the ISPC slopes are expected to have opposite signs for Congruent and Incongruent trials. 

5. If there are no impediments to this, could the authors please share the raw data of the experiment? I could not find them on OSF.

R: Raw files and matlab scripts to compute the variables of interest have been added to OSF 

Formatting and language

Line 22: Relate TO

Line 64: To THIS aim

Line 84-85: I think the authors mean “in the EMOTIONAL color-word Stroop task”. Also, I would probably call it emotional word-color Stroop, as it is the word to be emotional (not the color).

Line 95-96: “… to investigate whether the processing of emotions interacts with cognitive control and AT WHICH LEVEL”. This sentence may suggest that the authors intend to investigate the impact of emotional processing on control operating on different cognitive representations (e.g., conflict at the response level, or at the task level). I think “mode” would be avoid confusion here.

Line 119: Perhaps say “EMOTIONAL neutral” stimuli to stress that we’re not talking about neutral Stroop stimuli.

Line 134: Instead of “steal” perhaps use another word? Withhold? Employ? Occupy?

Line 146: Maybe without “therefore”?

Line 157: Maybe “Ethics” instead of “Ethical”?

Line 184: stroop with capital S.

Line 274: “At the end of the we performed”. Missing “practice phase”?

Line 319: DID not analyze (without d)

Line 393: “A second implication of our findings … “ is followed by the description of the findings themselves. I would remove the implication part.

R: We thank the reviewer. The manuscript has been revised accordingly to incorporate the suggested changes.

References.

Algom, D., Chajut, E., & Lev, S. (2004). A rational look at the emotional stroop phenomenon: a generic slowdown, not a stroop effect. Journal of experimental psychology: General, 133(3), 323.

Barr, D. J., Levy, R., Scheepers, C., & Tily, H. J. (2013). Random effects structure for confirmatory hypothesis testing: Keep it maximal. Journal of memory and language, 68(3), 255-278.

Bugg, J. M. (2012). Dissociating levels of cognitive control: The case of Stroop interference. Current Directions in Psychological Science, 21(5), 302-309.

Bugg, J. M. (2017). Context, conflict, and control. The Wiley handbook of cognitive control, 79-96.

Bugg, J. M., & Hutchison, K. A. (2013). Converging evidence for control of color–word Stroop interference at the item level. Journal of Experimental Psychology: Human Perception and Performance, 39(2), 433.

De Houwer, J. (2003). On the role of stimulus-response and stimulus-stimulus compatibility in the Stroop effect. Memory & Cognition, 31, 353-359.

De Pisapia, N., & Braver, T. S. (2006). A model of dual control mechanisms through anterior cingulate and prefrontal cortex interactions. Neurocomputing, 69(10-12), 1322-1326.

Tafuro, A., Vallesi, A., & Ambrosini, E. (2020). Cognitive brakes in interference resolution: A mouse-tracking and EEG co-registration study. Cortex, 133, 188-200.

 

Reviewer #2:

The authors performed an emotional priming Stroop test with list-wide (LWPC) and item-specific (ISPC) proportion congruence simultaneously manipulated. Findings suggest that sad priming led to behavioral costs only in trials with low proactive and reactive cognitive control. Proactive and reactive cognitive control can overcome the emotional interference of priming stimuli.

Overall, the work is of particular interest and bring important new insight in how emotion perception and cognitive control interact to shape behavior. The study is well-designed and the manuscript is generally well written with some issues presented below to be resolved:

R: We thank the reviewer for her/his positive evaluation and useful comments.

Line 28: In the abstract, low proactive and reactive cognitive control was mentioned when describing the results (but not methods) of the study. Consider adding some details in the sentences on methods to indicate how cognitive control was manipulated.

R: The description of the manipulation of proactive and reactive control modes was included in the abstract before describing the results: “To manipulate the requirement of both proactive and reactive control, the proportion of congruent trials (PC) was varied at the list-wide (LWPC) and item-specific (ISPC) levels, respectively”.

Line 78-80: The parenthesis “for our aim” can be removed in the sentence. This is same for other parentheses like “,then” (line 241), and “, more specifically” (line 65).

Line 94: The sentence should read “we employed an emotional priming spatial Stroop task to investigate ….” (the comma can be deleted).

R: Done. We thank the reviewer for these suggestions.

Line 121: Please define what is directional hypothesis since some readers may not be familiar with it.

R: In response to the reviewer's request, the revised version of the manuscript now states (lines 121-122): “Accordingly, it is challenging to formulate unique directional hypotheses that specify the expected relationship between emotions and cognitive control.” 

Line 194-200: The material list can be moved to appendix so as not to distract the reader.

R: The list has been moved to S1 Appendix.

Line 237-240: I’m a bit confused by seeing figure 2. How are blocks 1 and 5 (Also block 2 vs. block 4, block 3 vs. block 6) different? How are LWPC and ISPC orthogonalized?

R: List-wide probabilities were the same between blocks 1 and 5, 2 and 4, and 3 and 6. However, due to the variation in the presentation order, trial-wise probabilities differed and were balanced. As illustrated in Figure 3, during the initial stage of Block 1, trial-wise LWPC commenced at .5 (matching the LPWC of .5 during the practice block), progressively increased throughout the block, declined in Block 2, and returned to .5 in Block 3. The opposite trajectory was observed in Blocks 4-6. This aspect is now mentioned in the revised manuscript (lines 237-238). Concerning the relationship between LWPC and ISPC, we thank the reviewer for the question, as it provided us with an opportunity to clarify this particular point. The revised version of this section in the manuscript now states as follows (lines 238-245): “Since the effect of PC manipulations, especially of ISPC, has been challenged by associative theoretical frameworks (41–43), the number of trials for each combination of target direction and location was determined in such a way to orthogonalize the block-wise LWPC and ISPC with respect to probability of response given a stimulus location (PRS; also known as contingency). Moreover, we tried to lower the correlation between block-wise LWPC and ISPC as much as possible by carefully varying the occurrences of the different condition combinations and using ISPC values as different as possible compared to the LWPC values of each block.”

The revised text now additionally reports the shared variance between trial-wise LWPC-ISPC, LWPC-PRS, and ISPC-PRS (lines 276-277).

Line 249: What is the difference between the first and second “target congruency”. Please give specifications.

R: The first target congruency relates to the probability of congruency regardless of the target's position (i.e., LWPC). The second target congruency specifically considers one position at a time, representing P(Congruency|Position) (i.e., ISPC). To provide clarity, the revised sentence now reads as follows (lines 262-264): "Here, the probability of target congruency for trial-wise LPWC, the probability of target congruency at each specific location for trial-wise ISPC, and the probability of target direction at each specific location for trial-wise PRS."

P275: What was the purpose of the manipulation check to compare sadness with other emotions without neutrality? Please make justifications.

R: It was to verify that our experimental manipulation (i.e., presenting sad faces as emotional primings) was indeed effective in specifically evoking sadness in our participants (see lines 290-295).

Line 277: Please specify the p values apart from t values.

R: Done

Line 314: When reporting the results of linear mixed-effects models, consider providing the F values/χ2 for model comparisons as a justification for the final model (at least in supplemental materials).

R: The χ2 value for the log-likelihood ratio test between the full model and the final model is reported in the revised manuscript.

Line 410: It should be cautious to draw a conclusion that emotion processing affects processing stages other than cognitive control since other processing stages were not measured or manipulated in the current study.

R: In response to the second minor comment from Reviewer 1, this sentence has been amended as follows in the revised manuscript (lines 435-436): “Overall, taking our results into account, it seems that emotional interference affects cognitive processes when the level of cognitive control is negligible.” We hope that the revised sentence better conveys our line of reasoning: Emotional priming had an impact on certain cognitive processes, as indicated by the longer iRTs. This impact was observable only when the level of cognitive control was minimal.

Line 415: The sentences should read “First, our conclusions are limited to sadness. Thus, further studies are needed to generalize them to emotions with opposite valence or higher arousal.”

R: Done.

Line 414-431: This study can also be extended to cross-channel and cross-modal settings. Future studies can further explore the potential individual differences (e.g., gender differences) and task-related differences in auditory or audiovisual emotional Stroop priming effects. See references below:

1) Filippi, P., Ocklenburg, S., Bowling, D. L., Heege, L., Güntürkün, O., Newen, A., & de Boer, B. (2017). More than words (and faces): evidence for a Stroop effect of prosody in emotion word processing. Cognition and Emotion, 31(5), 879-891.

2) Lin, Y., Ding, H., & Zhang, Y. (2021a). Gender differences in identifying facial, prosodic, and semantic emotions show category- and channel-specific effects mediated by encoder's gender. Journal of Speech, Language & Hearing Research, 64(8), 2941-2955.

3) Lin, Y., Ding, H., & Zhang, Y. (2021b). Unisensory and multisensory Stroop effects modulate gender differences in verbal and nonverbal emotion perception. Journal of Speech, Language, and Hearing Research, 64(11), 4439-4457.

R: We thank the reviewer for this suggestion, which has been incorporated into the revised manuscript (lines 452-455).

---

## [Decision Letter · Decision Letter 1]

13 Nov 2023

On the relationship between emotions and cognitive control: Evidence from an observational study on emotional priming Stroop task

PONE-D-23-12904R1

Dear Dr. Vallesi,

We’re pleased to inform you that your manuscript has been judged scientifically suitable for publication and will be formally accepted for publication once it meets all outstanding technical requirements.

Kind regards,

Valerio Manippa

Academic Editor

PLOS ONE

Additional Editor Comments (optional):

Reviewers' comments:

Reviewer's Responses to Questions

**Comments to the Author**

1. If the authors have adequately addressed your comments raised in a previous round of review and you feel that this manuscript is now acceptable for publication, you may indicate that here to bypass the “Comments to the Author” section, enter your conflict of interest statement in the “Confidential to Editor” section, and submit your "Accept" recommendation.

Reviewer #1: All comments have been addressed

Reviewer #2: All comments have been addressed

2. Is the manuscript technically sound, and do the data support the conclusions?

Reviewer #1: Yes

Reviewer #2: Yes

3. Has the statistical analysis been performed appropriately and rigorously? 

Reviewer #1: Yes

Reviewer #2: Yes

4. Have the authors made all data underlying the findings in their manuscript fully available?

Reviewer #1: Yes

Reviewer #2: Yes

5. Is the manuscript presented in an intelligible fashion and written in standard English?

Reviewer #1: Yes

Reviewer #2: No

6. Review Comments to the Author

Reviewer #1: I appreciate the time and effort the Authors have invested in providing detailed responses to my queries. My concerns were addressed and I believe that the added clarifications have strengthened the manuscript.

Reviewer #2: The manuscript has been greatly improved. All my previous comments are addressed. Congratulations on the interesting findings and well-written manuscript!

7. PLOS authors have the option to publish the peer review history of their article (what does this mean?). If published, this will include your full peer review and any attached files.

Reviewer #1: **Yes: **Luca Moretti

Reviewer #2: No

---

## [Editor Report · Acceptance letter]

15 Nov 2023

PONE-D-23-12904R1 

On the relationship between emotions and cognitive control: Evidence from an observational study on emotional priming Stroop task 

Dear Dr. Vallesi:

I'm pleased to inform you that your manuscript has been deemed suitable for publication in PLOS ONE. Congratulations! Your manuscript is now with our production department. 

Kind regards, 

on behalf of

Dr. Valerio Manippa 

Academic Editor

PLOS ONE